# Isolation of Hepatic and Adipose-Tissue-Derived Extracellular Vesicles Using Density Gradient Separation and Size Exclusion Chromatography

**DOI:** 10.3390/ijms241612704

**Published:** 2023-08-11

**Authors:** Juan Alfonso Martínez-Greene, Margarita Gómez-Chavarín, María del Pilar Ramos-Godínez, Eduardo Martínez-Martínez

**Affiliations:** 1Laboratory of Cell Communication and Extracellular Vesicles, National Institute of Genomic Medicine (INMEGEN), Mexico City 14610, Mexico; martinezgreene@ie-freiburg.mpg.de; 2Physiology Department, School of Medicine, National Autonomous University of Mexico, Mexico City 04510, Mexico; margaritachavarin@gmail.com; 3Electron Microscopy Laboratory, National Institute of Cancer, Mexico City 14080, Mexico; pilyrg@gmail.com

**Keywords:** extracellular vesicles, tissue-derived extracellular vesicles, liver, adipose tissue, density gradient, size exclusion chromatography, exosomes, electron microscopy

## Abstract

In recent years, the study of extracellular vesicles (EVs) in the context of various diseases has dramatically increased due to their diagnostic and therapeutic potential. Typically, EVs are isolated in vitro from the cell culture of primary cells or cell lines or from bodily fluids. However, these cell culture methods do not represent the whole complexity of an in vivo microenvironment, and bodily fluids contain a high heterogeneous population of vesicles since they originate from different tissues. This highlights the need to develop new methods to isolate EVs directly from tissue samples. In the present study, we established a protocol for isolating EVs from hepatic and adipose tissue of mice, using a combination of ultracentrifugation and iodixanol-sucrose density gradient separation. EV isolation was confirmed with EV protein marker enrichment in Western blot assays, total protein quantification, and transmission electron microscopy. Regarding the liver tissue, we additionally implemented size exclusion chromatography (SEC) to further increase the purity grade of the EVs. The successful isolation of EVs from tissue samples will allow us to uncover a more precise molecular composition and functions, as well as their role in intercellular communication in an in vivo microenvironment.

## 1. Introduction

Intercellular communication is a fundamental process for the homeostatic maintenance of cells, tissues, and whole organisms. There is a wide array of biochemical molecules and interactions by which cells communicate with each other, with one of them being the secretion of extracellular vesicles (EVs) into the extracellular milieu and its subsequent reception by distant cells. EVs are particles secreted into the extracellular space, which contain a particular set of proteins, lipids, and nucleic acids and may be able to modify the metabolism of recipient cells [1,2]. EVs can be separated into three main groups: apoptotic bodies, microvesicles, and exosomes. Usually, EVs are collected and studied from cell culture media of cell lines or tissue explants in vitro [3,4].

Despite the common use of cell culture to obtain EVs, there is an increasing concern that these conditions may not reflect the physiological state of EVs. It has been demonstrated that the material and the surface topology of a cell culture dish/flask may alter key cell properties that could eventually modify the composition of EVs and its subsequent function [5,6]. Moreover, the components used in cell culture growth media to maintain cell proliferation may modify cell metabolism. Careful experimental evaluation of specific growth media for specific cell types is needed to determine which is best suitable for cell culturing [7]. Failure to do so may result in significant alterations in cell metabolism, proliferation, and gene expression. Taken together, these factors may lead to altered results of EV composition and function that may not be suitable to be associated with in vivo processes.

To counteract these caveats, there is a growing interest in obtaining EVs directly from tissue samples [4,8,9,10,11,12]. It is expected that tissue-derived EVs reflect their native composition and function more precisely than cell culture methods. In 2012, the first method to isolate EVs from the cerebral tissue of mice and humans appeared [10]. Since then, several EV isolation protocols have been developed to obtain EVs from cerebral, adipose, and tumoral tissues, both from animal and human samples [9,11,13,14]. However, there is still a lack of studies that isolate vesicles from other tissue sources, such as hepatic, renal, cardiac, and muscular tissues. Moreover, proteomic studies from brain tissue-derived EVs suggest that these preparations may co-isolate a considerable fraction of non-vesicular proteins [9,11,14]. There is a need to improve isolation methods and characterize vesicles from unexplored tissues.

Currently, as there is no standardized method to obtain EVs from different tissue samples, it is challenging to compare results between studies from different laboratories. A combination of different isolation methods has been shown to increase yield and purity by preserving EV characteristics [15,16]. Here, we present a workflow that combines ultracentrifugation and density gradient separation to obtain EVs from hepatic tissue and adipose tissue. We also evaluated the effects of different enzymes to release EVs from tissues. Finally, we added a size exclusion chromatography (SEC) step to the isolation protocol of hepatic-tissue-derived EVs to increase sample purity. Our results indicate that the selection of enzymatic treatment will depend on the tissue studied and that the addition of the SEC step helped to eliminate non-vesicular proteins.

## 2. Results

### 2.1. Isolation of Extracellular Vesicles from Tissue

We isolated EVs from frozen tissue according to the workflow depicted in Figure 1 which included steps of differential centrifugation and density gradient separation of the EV-enriched pellet. In the case of the isolation of vesicles from hepatic tissue, we digested the tissue with papain. The highest amount of protein was found in the first five fractions of the gradient. Fraction 3 presented the highest protein content, with almost five times more protein than fractions 2 and 4 (Figure 2a). The vesicle markers annexin a2 (Anxa2), annexin a5 (Anxa5), and lactadherin (Mfg-e8) were distributed in the first five fractions of the gradient (Figure 2b). We also tested the action of type II collagenase with hepatic tissue. The total protein obtained from the fractions of the density gradient was similar to the amount of protein obtained with papain (Figure 2c). However, the enzyme collagenase caused cell rupture in the hepatic tissue as evidenced by the presence of calnexin in fractions 3 to 5 of the gradient. Vesicle markers such as Mfg-e8 and Anna2 were still present in the first fractions of collagenase-digested tissue, while Anna5 was only detectable in fraction 3 (Figure 2d). Furthermore, tumor susceptibility gene protein 101 (Tsg101) was detected in fractions 2 to 5 of the gradient, contrary to the use of papain, where it was not detected. Finally, Cd81 and Cd63 (not shown) could not be detected either with papain nor collagenase (Figure 2b,d). To evaluate whether there was protein contamination such as lipoproteins, we determined the presence of Apob in our preparations. There was more co-isolation of apolipoprotein B (Apob) in tissue digested with collagenase than in tissue digested with papain (Figure 2b,d). Apob was found from gradient fraction 1 to gradient fraction 5 in both cases.

To determine if part of the isolated proteins came from blood components, we also isolated vesicles from the hepatic tissue of perfused mice (Appendix A). The total protein recovered was six times less than the total protein obtained from the non-perfused tissue. We were only able to detect Anxa2 in fraction 1 of the gradient (Appendix A). The contamination marker calnexin was absent from the fractions.

To evaluate the versatility of the isolation method, we decided to isolate vesicles from 0.5 g of mesenteric adipose tissue (MAT) from mice (Figure 3a). Initially, we used the papain to disaggregate the tissue. However, this enzyme disintegrated the tissue, causing cell rupture, which was confirmed by the presence of calnexin in the fractions of the gradient (Figure 3b). After digesting the adipose tissue with type II collagenase, most of the protein was also concentrated in the first four fractions of the density gradient. Interestingly, calnexin was not detected with type II collagenase. The vesicle markers Cd81, Cd63, and Anxa2 were enriched in the first four fractions of the gradient, while anxa5 was distributed along the gradient. The markers Mfg-e8 (not shown) and Tsg101 were not detected in the fractions from the adipose tissue (Figure 3c). For positive control of EV markers, we obtained MAT enriched homogenates of cytosolic protein and membrane protein (Figure 3c).

### 2.2. Significant Removal of Non-Vesicle Material by SEC

The discrepancy between the total protein isolated in the gradient fractions from the liver tissue and the signal intensity from the vesicle markers led us to believe that a great portion of the isolated protein was of non-vesicle origin. Therefore, we applied a third isolation step to try to increase the purity of the isolated vesicles. The fractions obtained from the density gradient were paired up, and then size exclusion chromatography was performed on each paired sample, and 30 fractions were collected from each sample (Figure 1). By determining the relative protein content, we identified two peaks of protein: one between fractions 5 to 10 and a second one between fractions 13 to 28. Since the vesicles are usually expected in the first fractions of the chromatography [17], we concentrated fractions 5 to 10 by ultracentrifugation, from each paired sample (Figure 1). Then, we determined the total protein recovered, vesicle protein markers, and particle morphology (Figure 4). The total protein quantification showed an almost 50% decrease in protein concentration in the SEC fractions of paired sample 1–2 compared to the same fractions before chromatography separation (Figure 2a). Also, we found a 75% protein decrease in paired samples 3/4 and 5/6, and, finally, there were no significant changes in the protein content of paired samples 7/8 and 9/10 (Figure 4b). The Western blot assays revealed that markers Mfg-e8 and Anxa2 were still present in the paired samples 1/2 and 3/4, just as previously seen in the fractions of the gradient. In paired sample 5/6, only MFG-E8 was detected. Surprisingly, the CD81 marker was detected in paired samples 1/2 and 3/4 after chromatography separation, which was previously not detected in the fractions of the density gradient. Furthermore, we detected a 10–15 KDa protein band using the CD81 antibody. In contrast, it was not possible to detect vesicle markers in paired samples 7/8 and 9/10 (Figure 4c). Apob was mainly found in F1-2 and F3-4 (Figure 4c).

To corroborate the presence of vesicular structures we obtained transmission electron microscope (TEM) images before and after SEC. After the density gradient, we observed the presence of vesicles in paired fractions 1-2 and 3-4, while paired fractions 5-6 had a heterogeneous population of particles with a minimal presence of vesicles. Finally, paired fractions 7-8 and 9-10 did not have any visible vesicles present (Figure 5a). The addition of SEC as a third isolation step did not modify the distribution of the EVs but eliminated proteinaceous material throughout the paired fractions (Figure 5b).

## 3. Discussion

Recently, interest in the isolation of EVs from tissue samples has greatly increased because it has been possible to obtain vesicles that have been secreted into the interstitial space within tissues and analyze their molecular composition. Some studies have shown the isolation of vesicles from different organs and tissues such as the brain, adipose tissue, and tumors [4,9,11,14,18,19]. In this work, we showed that the addition of a SEC step after density gradient ultracentrifugation still eliminates non-vesicular proteins. This suggests that there is co-isolation of protein aggregates after enzymatic digestion and highlights the need of including different strategies to assure EV purity.

In all cases, the isolation of tissue-derived EVs involves the combination of an enzymatic digestion reaction with several steps of sample purification. Since these methods to isolate vesicles directly from tissues are relatively new [8], it is complex to directly compare the results from different laboratories. For example, just in the case of the brain tissue, several enzymes and solutions have been proposed for their use in the tissue disaggregation step [9,10,11,13,18]. In our case, we decided to use papain (10 units/mL) or type II collagenase (0.15%) dissolved in 1x PBS to facilitate protocol replication [20].

The enzymatic digestion experiments revealed that the choice of the enzyme for tissue disaggregation is critical for successful EV isolation while avoiding cell damage. In the case of the hepatic tissue, disaggregation using type II collagenase caused cell rupture, determined by the presence of calnexin in the density gradient fractions, while it was undetectable using papain. In contrast, the use of collagenase to digest adipose tissue allowed for proper disaggregation without causing cell rupture. Papain belongs to the cysteine hydrolase family of enzymes, which break peptide bonds, and thus have a wide range of substrate specificity [21]. Conversely, collagenases are endopeptidases that degrade collagen fibers at their triple-helix section. Specifically, type II collagenase refers to a mixture of enzymes, mainly enriched in collagenase, where caseinase, trypsin and clostripain are also present, with the latter being particularly enriched in this mixture [21]. Both papain and type II collagenase have been recommended to disaggregate hepatic and adipose tissues [21]. It is important to note that using collagenase for tissue digestion does not significantly alter cell membrane proteins [22]. While using papain has not shown to significantly decrease cell viability [23], it is necessary in future studies to determine the impact of papain in the integrity of membrane proteins of EVs [24]. The differences found in the tissue digestion between hepatic and adipose tissues can be due to several factors, such as extracellular matrix composition, mechanical desegregation, and sample freshness, among others. It has been reported that vesicle isolation success is dependent on the orientation and deepness of the incisions made to the tissue [11]. Also, one freeze–thaw cycle of brain tissue has not been shown to alter the enzymatic digestion results [10], although this has not been demonstrated in hepatic nor adipose tissues. Further experiments are required to evaluate all of the aforementioned factors that could affect the enzymatic digestion of the tissue. With our data, we cannot discard that enzymatic digestion used to release EVs could affect the structure of membrane proteins, thus limiting functional studies.

The Western blot assays revealed high heterogeneity between the presence and intensity of EV protein markers of vesicles obtained from hepatic and adipose tissues. For example, the hepatic-derived vesicles showed the presence of Mfg-e8, Anxa2, and Anxa5, while the mesenteric-adipose-tissue-derived vesicles showed the presence of EV protein markers Cd81, Cd63, Anxa2 and Anxa5. Mfg-e8 was not detected in the adipose-tissue-derived vesicles, but we observed a strong signal in vesicle-enriched fractions from the hepatic tissue. Annexins and Mfg-e8 are usually found in high concentrations in cell membranes and the blood, since they are involved in processes such as blood coagulation, membrane repair, fibrinolysis, and angiogenesis. It has been shown that the presence of these proteins in the blood is not in soluble form but rather strongly associated with vesicles [25]. Annexins are usually found in the cytoplasm, so their presence in the blood strongly suggests a vesicle-associated secretion pathway. Mfg-e8 can recognize phosphatidylserine on cell membranes, which is why its presence has also been associated with vesicle transport [26,27]. Cd81, which is considered a canonical EV protein marker, was detected in the hepatic vesicles only after adding the SEC isolation step. This could be due to two main reasons: (1) hepatic vesicles present low enrichment of this protein or (2) the sample still has high presence of soluble proteins that mask this protein signal, even after being purified by the SEC step. Cd63, which is another canonical EV marker, showed strong enrichment in adipose-derived vesicles. The differential levels of the analyzed vesicle markers seem to be related to the expression levels of each protein in different tissues. For example, we observed that Cd63 levels are especially high in the kidney and adipose tissue, while Mfg-e8 is particularly high in the hepatic tissue (Appendix A). Other groups have characterized different subpopulations of vesicles extracted from different melanoma tissues and found that there is variation in the presence and enrichment of EV markers not only between the same tissues but also between vesicle subpopulations [4].

Some studies have opted to previously perfuse tissues before vesicle isolation (in animal tissues) to reduce possible contamination coming from blood and increase proteomics resolution [14]. However, it is challenging to obtain perfused human tissue. Therefore, it is necessary to develop novel protocols to be able to purify vesicles directly from tissues without perfusion. We compared the effects of hepatic tissue perfusion on vesicle isolation using our protocol. The total protein recovered decreased 30% on perfused tissue, compared to non-perfused tissue. Huang et al. evaluated the effects of perfusion on mice and macaque’s cerebral tissues and found no significant differences in their overall vesicle isolation results. In our case, perfusion did significantly alter mean total protein recovery. Also, Mfg-e8 and Anxa5 signals disappeared, while the Anxa2 signal was restricted to fractions 1 and 2 of the density gradient. This suggests that the previous signal detected in the non-perfused hepatic tissues could come from the blood, rather than the interstitial space. It is known that the vascular system has a fluid and molecule exchange with the interstitial space from tissues through a process called capillary exchange [28]. This exchange is dependent on several factors, such as hydrostatic pressure, osmotic pressure, and the spatial position of the capillaries. It has been shown that, by increasing capillary flow, the capillary permeability surface area also increases [29]. Therefore, it is possible that, when perfusing the tissues, there could be a loss of interstitial liquid because of the increase in capillary flow and, thus, a loss of vesicular material present in the interstitial space. Further experiments are necessary to evaluate and determine what proportion of the isolated vesicles come from the interstitial space or the blood within tissues. Regarding soluble protein contaminants, our results showed that there is co-isolation of Apob lipoprotein even after SEC, mainly explained by the size overlapping. Recently, it has been shown that Apob content in EV preparations can be further reduced by using multimodal chromatography which combines resins with different capacities [30].

The analysis of tissue-derived EVs is a promising field for understanding the progression of different diseases in which there is an animal model or there is the availability of clinical samples. An advantage of the molecular characterization of tissue-derived EVs is that it allows to capture the biochemical composition of EVs that may not be easily characterized in body fluids. Moreover, sampling overtime of a disease model will also benefit our understanding of EVs’ role in cell–cell communication. Thus, it is imperative to generate easily reproducible protocols for a straightforward comparison of the molecular content of EVs from different developmental ages and disease states. In this work, we showed that the classical protocol for tissue-derived EVs, based on density gradient ultracentrifugation, could benefit from the additional step of purification. The incorporation of immunocapture strategies or novel ways of nanoparticle fractionation would also increase the resolution of proteomic studies in the near future [16].

## 4. Materials and Methods

### 4.1. Animals and Organ Extraction

For organ extraction, we used CD1 male and female mice which were anesthetized with an intraperitoneal injection of 0.1 mL of sodium pentobarbital. The mice were sacrificed by decapitation. Then, we dissected the liver and mesenteric adipose tissues from each mouse. The samples were immediately placed in 2 mL tubes, immersed in an isopentane bath chilled with dry ice, and stored at −80 °C until further use. To evaluate the contribution of blood components to the EV preparations, we perfused the mice by cardiac puncture using 0.9% NaCl. The animals were obtained from the animal facility of the Instituto de Investigaciones Biomédicas (IIB), Universidad Nacional Autónoma de México. Animal handling and the experimental protocols were revised and approved by the Animal Rights Committee at IIB, UNAM.

To determine the levels of vesicle markers in different mouse tissues, we enriched integral membrane proteins with a MEM-PER Plus Membrane Protein Extraction Kit (Thermo Fisher Scientific, Rockford, IL, USA). The frozen tissue samples were processed according to the manufacturer’s specifications, which allowed for the obtention of a fraction of cytosolic proteins and other fraction of membrane proteins. The samples were supplemented with protease inhibitors and EDTA (Halt Protease Inhibitor Single-Use, Thermo Fisher Scientific).

### 4.2. Cell Culture

As a positive control, we used the hepatic mouse cell line AML12 for EV isolation. Cells were cultivated in DMEM/F12 medium supplemented with 10% fetal bovine serum, 1× insulin–transferrin–selenium, 40 ng/mL dexamethasone and penicillin/streptomycin. For the collection of conditioned media, we supplemented the culture media with 10% exosome-depleted fetal bovine serum [31]. After 48 h of culture, we collected the media and centrifuged them at 400 RCF for 10 min. The recovered supernatant was then centrifuged at 2000 RCF for 20 min. The final supernatant was recovered and stored at −80 °C until further use.

### 4.3. Isolation of Extracellular Vesicles

We ultracentrifuged the cell culture media at 118,000 RCF (40,000 RPM, k-factor 133.4) for 90 min in a fixed angle rotor (70Ti rotor, Beckman Coulter, Brea, CA, USA) to concentrate the EVs, as described previously [17]. The supernatant was discarded, and the pellet was recovered and resuspended in 2.8 mL of PBS. This sample was again ultracentrifuged at 118,000 RCF for 38 min (53,000 RPM, k-factor 54) at 4 °C in a fixed angle rotor (TLA 100.3, Beckman Coulter). We resuspended this pellet in 100 μL of RIPA buffer with protease inhibitors and EDTA.

For isolation of vesicles from the tissue samples, we used 0.5 g of each tissue. All samples were subjected to only one freeze–thaw cycle. The tissue was cut into small pieces using a razorblade and placed in a tube with 3.5 mL of papain dissolved in PBS (20 units/mL) or collagenase type II (0.15% *w*/*v*; Thermo Fisher Scientific, 17101015) depending on the tissue. The samples were incubated for 20 min or 30 min with the papain or collagenase solutions, respectively, at 37 °C with constant mixing. Enzymatic digestion was inactivated by adding 3.5 mL of PBS with 1× protease inhibitors and 1× EDTA (Halt™ Protease Inhibitor Cocktail, Thermo Fisher Scientific). The samples were then centrifuged at 400 RCF for 10 min to the remove cells. The supernatant was centrifuged at 2000 RCF for 20 min to remove apoptotic bodies and cell debris. The resultant supernatant was centrifuged at 10,000 RCF for 40 min. Finally, we filtered the supernatant using a 0.8 μm filter unit (Millipore) and then ultracentrifuged it at 118,000 RCF for 90 min (70Ti rotor, Beckman Coulter). The supernatant was discarded, and the pellet was resuspended in 2.8 mL of PBS and ultracentrifuged again at 118,000 RCF for 35 min (TLA100.3 rotor, Beckman Coulter). The supernatant was discarded and 2.5 mL of a 30% Optiprep (iodixanol)–sucrose gradient solution (d = 1.175 g/mL) was added to the pellet [32]. The mix was incubated overnight at 4 °C.

For gradient separation, a discontinuous gradient of Optiprep (iodixanol)–sucrose was employed, composed of 2.5 mL of 30% Optiprep (d = 1.175 g/mL), 1.3 mL of 20% Optiprep (d = 1.127 g/mL), and 1.2 mL of 10% Optiprep (d = 1.079 g/mL), with the sample placed at the bottom of the gradient (bottom loaded). The sample was ultracentrifuged at 250,000 RCF, 60 min with no brake (SW55 Ti rotor, Beckman Coulter, Brea, CA, USA). Ten fractions of 490 μL each were recovered, from top to bottom. Each fraction was washed with 1 mL of PBS and then ultracentrifuged at 118,000 RCF for 35 min (TLA 100.3 rotor, Beckman Coulter). Each fraction was resuspended in 60 μL of RIPA buffer, with protease inhibitors and EDTA or PBS, depending on further use.

For further purification of the hepatic-derived EVs, the recovered gradient fractions were resuspended in 500 μL of PBS per fraction, and grouped into pairs of fractions (1/2, 3/4, 5/6, 7/8, and 9/10), obtaining 1 mL per paired sample. Size exclusion chromatography was then performed using a Sepharose CL-2B column. Thirty fractions of 500 μL each were collected per paired sample, using a PBS/citrate 0.32% solution as eluent. Based on 280 nm absorbance measurements, we concentrated fractions 5 to 10 from each paired sample, which are vesicle-enriched fractions [17].

### 4.4. Transmission Electron Microscopy (TEM)

Vesicle isolation was performed as described in the Section 4.3 and TEM sample preparation as we described previously [17]. Once isolated, the vesicles were resuspended in 400 uL of PBS. The samples were concentrated in 0.5 mL Amicon 3 KDa filter tubes by centrifuging at 14,000 RCF up to a volume of 100 μL. Then, we exchanged PBS for a solution of paraformaldehyde/glutaraldehyde 2.5% (Electron Microscopy Sciences, Hatfield, PA, USA) The volume of the sample was reduced until it reached 100 μL and then we fixed the sample for 45 min. We placed 7 μL of the fixed sample over an ultrathin Formvar carbon grid and incubated it for 20 min. Finally, we performed 7 washes of 2 min each with distilled water and let the grids dry overnight at room temperature. The samples were counterstained using alcoholic uranyl acetate for 15 min and then washed twice with distilled water. The grids were imaged on a JEOL JEM-1010 electron microscope equipped with an AMT digital camera.

### 4.5. Western Blot

The following monoclonal antibodies were used for the experiments: Cd9 (Dilution (Dil.) 1:500, cat. # SC-13118, Santa Cruz Bitoechnology (SCBT), Dallas, TX, USA), Cd63 (Dil. 1:500, cat. # SC-5275, SCBT), Cd81 (Dil.1:500, cat. # SC-166029, SCBT), Annexin A5 (Dil. 1:2500, cat. # SC-74438, SCBT), Alix (Dil. 1:500, cat. # SC-53540, SCBT), TSG101 (Dil. 1:500, cat. # SC-7964, SCBT), MFG-E8 (Dil. 1:2500, cat. # SC-271574, SCBT), Cd63 (Dil. 1:500, cat. # ab217345, Abcam, Waltham, MA, USA), and Annexin A2 (Dil. 1:2000, cat. # ab178677, Abcam) as vesicle markers. We used goat anti-calnexin (Dil. 1:10,000, cat. # SC-6465, SCBT) and Apob (Dil. 1:1000, cat. # sc-393636, SCBT) as a contamination marker. The secondary reagents included anti-IgG kappa-binding protein (cat. # SC-516102, SCBT) to detect primary mouse antibodies, anti-rabbit-HRP (cat. # SC-2357, SCBT), and anti-goat-HRP (cat. # SC-2020, SCBT).

## 5. Conclusions

Our results indicate that enzyme selection for each tissue is critical to achieve precise extracellular matrix disaggregation to extract the vesicles, without causing cell rupture. Therefore, we determined that papain is the best option for hepatic tissue while type II collagenase is the best option for adipose tissue. We found that tissue perfusion drastically alters how much total protein is obtained, as well as the presence and enrichment of vesicle protein markers in the isolated material from the tissues.

With these results, we determined that the addition of size exclusion chromatography as a third isolation step removes a significant quantity of soluble proteins from the samples, without altering vesicle integrity.

## Figures and Tables

**Figure 1 ijms-24-12704-f001:**
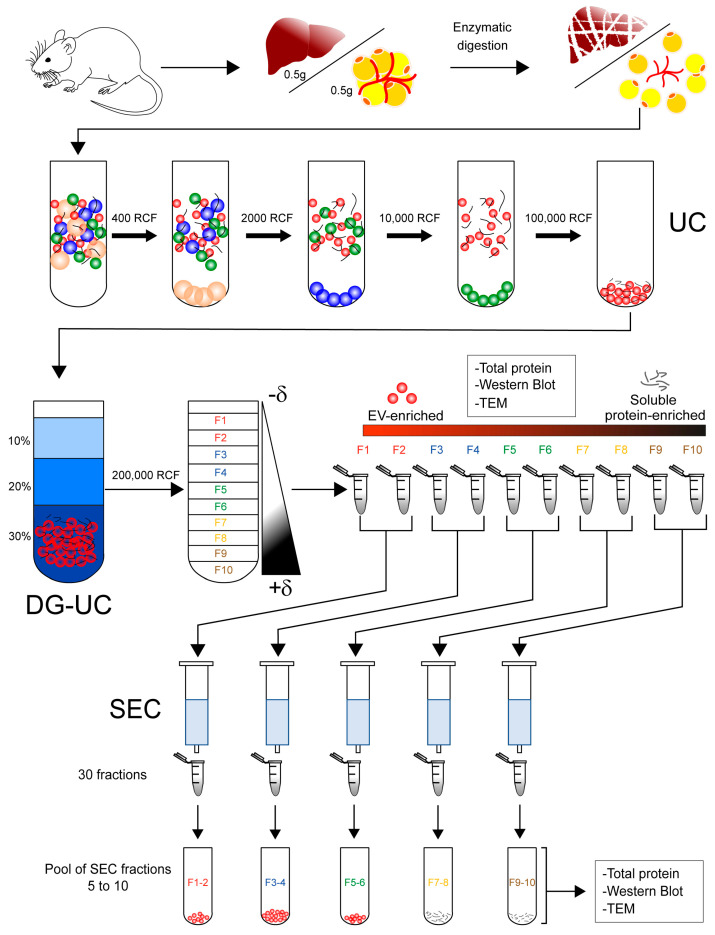
Isolation protocol to obtain tissue-derived extracellular vesicles from different tissue sources. After the digestion of hepatic and adipose tissue, the homogenate was subjected to differential centrifugation to obtain an EV-enriched pellet. This sample was loaded at the bottom of an Optiprep–sucrose density gradient and ultracentrifuged for 60 min. To increase the EV purity, the gradient fractions were subjected to size exclusion chromatography, and pooled fractions of the chromatography were analyzed for the presence of EVs. UC = ultracentrifugation, DG-UC = density gradient ultracentrifugation, SEC = size exclusion chromatography, and TEM = transmission electron microscopy.

**Figure 2 ijms-24-12704-f002:**
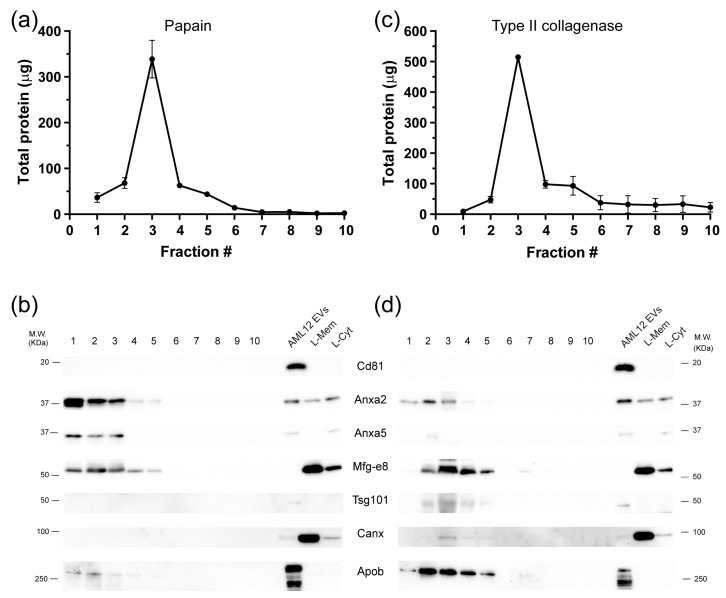
Comparison of the effect of papain and collagenase digestion of hepatic tissue on EV output. (**a**) Total protein per fraction obtained after the DG-UC of hepatic tissue digested with papain. (**b**) An equal volume of the resulting fractions of papain digested tissue was analyzed for the presence of EV markers by Western blot. (**c**) Total protein per fraction obtained after the DG-UC of hepatic tissue digested with type II collagenase. (**d**) EV markers present in fractions obtained by the DG-UC of hepatic tissue digested with collagenase. With collagenase there was a decrease in the EV markers’ signal, especially from fractions 1 to 3, and there was presence of calnexin, indicating cell rupture. To determine the presence of lipoprotein contaminants, we evaluated the presence of Apob. Note that papain digestion co-isolated less Apob than collagenase treatment. As a positive control, we isolated EVs from the mouse cell line AML2 and prepared hepatic homogenates enriched in membrane proteins (L-Mem) and cytosolic proteins (L-Cyt).

**Figure 3 ijms-24-12704-f003:**
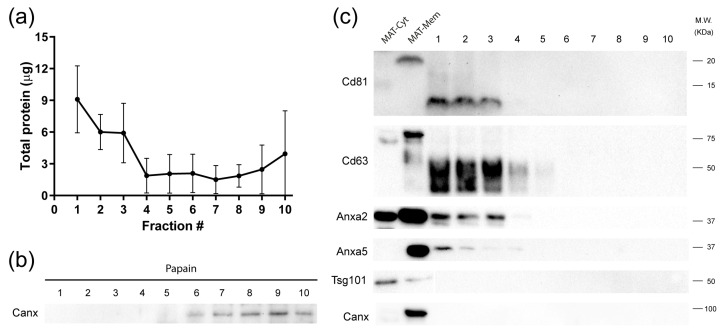
EVs derived from mouse mesenteric adipose tissue (MAT). (**a**) Total protein per fraction obtained after the DG-UC of mesenteric adipose tissue digested with type II collagenase. (**b**) The use of papain to digest adipose tissue resulted in cell rupture as indicated by the presence of Canx in density gradient fractions. (**c**) An equal volume of the resulting fractions of collagenase-digested adipose tissue was analyzed for the presence of EV markers by Western blot.

**Figure 4 ijms-24-12704-f004:**
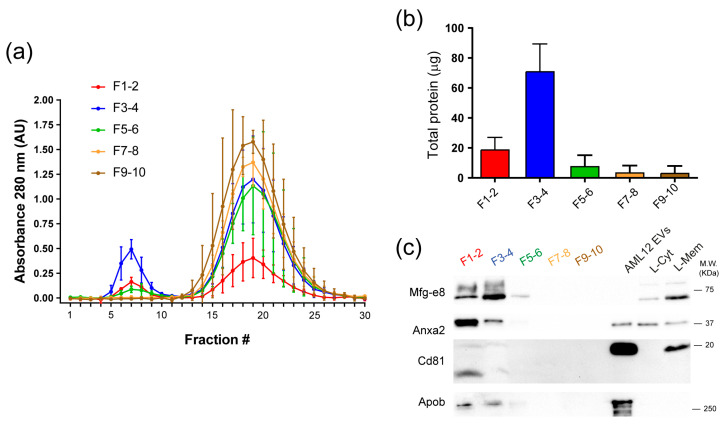
Additional purification of EVs derived from hepatic tissue by size exclusion chromatography (SEC). (**a**) Relative protein content in each SEC fraction. The pair of fractions obtained from density gradient ultracentrifugation were loaded into a Sepharose CL-2B column, and the absorbance at 280 nm was measured for each fraction. Note that F1-2, F3-4, and F5-6 showed protein content in SEC fractions 5 to 10, which correspond to the EV fractions. (**b**) The total protein content of pooled SEC fractions 5 to 10. (**c**) Western blot analysis of pooled SEC fractions for Mfg-e8, Anxa2, and Cd81, Apob. Note the appearance of an additional band (10–15 KDa) positive for Cd81 in F1-2 and F3-4 as compared to the EVs derived from the AML12 cell line and L-Mem.

**Figure 5 ijms-24-12704-f005:**
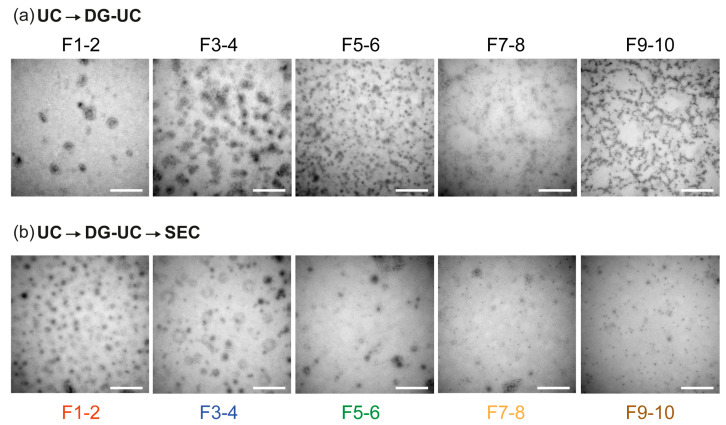
Photomicrographs of hepatic EVs obtained after density gradient ultracentrifugation and size exclusion chromatography. (**a**) Representative photomicrographs obtained from EV preparations after density gradient ultracentrifugation using the Optiprep–sucrose density gradient. To simplify the analysis, we paired the density gradient fractions as shown in Figure 1. (**b**) After SEC, fractions 5 to 10 of each paired fraction were concentrated and were processed for TEM analysis. Note that the addition of the SEC step reduced the extravesicular material. Scale bar = 500 nm.

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
