# Peer review of "Isolation of Hepatic and Adipose-Tissue-Derived Extracellular Vesicles Using Density Gradient Separation and Size Exclusion Chromatography"

_ijms, 2023, doi:10.3390/ijms241612704_

Round 1

Reviewer 1 Report

The authors developed a method to isolate extracellular vesicles (EVs) from liver or adipose tissues. Briefly, they used frozen tissues to be cut and digested with either papain or collagenase with shaking, and differential ultracentrifugation to concentrate EVs, which were further purified through iodixanol gradient ultracentrifugation. The purified EVs can be further purified through a size-exclusion chromatography approach. They measured the EV markers by Western blot and observed the morphology through transmission electron microscopy.

1.     My major concern is the authors used frozen tissues but not fresh tissues to isolate EVs. Upon freeze-thaw process, the tissues/cells may get ruptured, and this may also affect EV purity and quantity based on some literature. The authors need to explain the rationale to use frozen tissues but not fresh tissues. If the freeze-thawing process is required for the protocol, how many times of freeze-thaw were involved?

2.     Secondly, the authors need to explain why they used papain for tissue digestion. To my knowledge, collagenase is commonly used, while papain would make harm to the proteins associated with EVs.

3.     Because of the freeze-thaw process, the cells get ruptured, it is not avoidable to get cellular protein, for example, calnexin potentially contaminated with the EV samples. So the rupture of tissues/cells may be irrelevant to any proteinase application. In addition, it doesn’t mean that the results in Figure. 3a and 3b were not meaningful. I personally think Figures 3a and 3b implied that the cellular proteins were well separated from the EVs after the gradient ultracentrifugation, especially if the authors could detect the enrichment of EV markers in the first 3 fractions. Please provide such evidence of where the EV proteins were localized after the gradient centrifugation in Figure 3b.

4.     It would be better to label the raw Western blot images with molecular weight and put arrows to indicate the specific bands. Please also check especially the Tsg101 blots in suppl figure 2, I think the strong bands but not the weak bands just above were actually Tsg101.

5.     As you discussed in the discussion, serum marker like albumin should be measured in the purified sample after gradient centrifugation or SEC to better understand to what extent serum proteins were co-purified with the tissue EVs.

Minor issues:

1.     Please explain the meaning of “density gradient ultracentrifugation does not eliminate extravesicular material.” In line 187 on page 7.

2.     Typo in line 21 on page 1 “high purity EVs”.

English is acceptable. 

Reviewer 2 Report

Authors present nice paper on the isolation of EVs with multiple technique, however there are some points that need to be addressed before the paper could be accepted

Lines 74-74. The authors suggest a novel isolation method. However, there have been many cases where dgUC and SEC have been in combination, and it is often nessecity to use multiple techniques to obtain good quality of the isolate. Authors should get familiar with the literature and improve the background section. The referenced literature should also be up to date.

Find more on combining techniques for the isolation of EVs:  https://doi.org/10.1016/j.chroma.2020.461773

Authors should also comment advantages of this approach compared to advanced state of the art  subpopulation isolation systems:

https://doi.org/10.1021/acs.analchem.0c01986

Lines 75 and -76: What do authors think of the effect of these enzymes on tetraspanins of the EVs and other surface proteins? Why those would not be affected?

Lines 133-156: Analysis on the amount of lipoproteins present is missing.

lines 190-191: There are some methods that allow for comparison:  https://doi.org/10.1016/j.aca.2020.06.073

Lines 201-202, 210-211: Why would papain not break peptide bonds of EV proteins found on the surface e.g., tetraspanins? Please comment in text.

Lines 264-266: This should be done for this study, because both UC and SEC are known to be contaminated with lipoproteins and other proteins: https://doi.org/10.1016/j.chroma.2020.461773 

Overall OK.

Round 2

Reviewer 2 Report

The authors have corrected the manuscript according to the comments.